# Ultrasound Radiomics Nomogram to Diagnose Sub-Centimeter Thyroid Nodules Based on ACR TI-RADS

**DOI:** 10.3390/cancers14194826

**Published:** 2022-10-03

**Authors:** Wenwu Lu, Di Zhang, Yuzhi Zhang, Xiaoqin Qian, Cheng Qian, Yan Wei, Zicong Xia, Wenbo Ding, Xuejun Ni

**Affiliations:** 1Department of Medical Ultrasound, Affiliated Hospital of Nantong University, Medical School of Nantong University, Nantong 226001, China; 2Affiliated Hospital of Integration Chinese and Western Medicine with Nanjing University of Traditional Chinese Medicine, Nanjing 210023, China; 3Department of Ultrasound, Affiliated People’s Hospital of Jiangsu University, Zhenjiang 212050, China

**Keywords:** radiomics, thyroid nodules, thyroid imaging reporting and data system (TI-RADS), ultrasound, nomogram

## Abstract

**Simple Summary:**

Fine-needle aspiration or surgical resection is used to determine whether thyroid nodules (TNs) are benign or malignant, but there is a risk of overtreatment. Guidelines such as the American College of Radiology thyroid imaging reporting and data system consider follow-up for TNs < 1 cm, even with high-risk ultrasound features; therefore, for these TNs, an ultrasound radiomics nomogram was developed for noninvasive assessment of benign and malignant TNs. In addition, to assess the model’s diagnostic capabilities, the present study tested the model using an external validation group. The test results suggest that ultrasound radiomics is an effective, noninvasive method by which to diagnose noninvasive TNs.

**Abstract:**

The aim of the present study was to develop a radiomics nomogram to assess whether thyroid nodules (TNs) < 1 cm are benign or malignant. From March 2021 to March 2022, 156 patients were admitted to the Affiliated Hospital of Nantong University, and from September 2017 to March 2022, 116 patients were retrospectively collected from the Jiangsu Provincial Hospital of Integrated Traditional Chinese and Western Medicine. These patients were divided into a training group and an external test group. A radiomics nomogram was established using multivariate logistics regression analysis using the radiomics score and clinical data, including the ultrasound feature scoring terms from the thyroid imaging reporting and data system (TI-RADS). The radiomics nomogram incorporated the correlated predictors, and compared with the clinical model (training set AUC: 0.795; test set AUC: 0.783) and radiomics model (training set AUC: 0.774; test set AUC: 0.740), had better discrimination performance and correction effects in both the training set (AUC: 0.866) and the test set (AUC: 0.866). Both the decision curve analysis and clinical impact curve showed that the nomogram had a high clinical application value. The nomogram constructed based on TI-RADS and radiomics features had good results in predicting and distinguishing benign and malignant TNs < 1 cm.

## 1. Introduction

The occurrence of thyroid nodules (TNs) has increased in recent years in several countries throughout the world; however, this increase could be attributed to the availability of more accurate diagnostic and detection techniques [1]. The prevalence of TNs is ~30–67%, comprising mainly malignant and benign nodules [2], with ~7–15% being thyroid cancer. Except for those individuals with higher risk factors, screening for thyroid cancer is recommended. Routine screening for malignancy, such as fine-needle aspiration (FNA), is not recommended, especially for TNs < 1 cm that were conservatively observed [3]; however, 52.8% of patients already had cervical lymph node metastasis while having papillary thyroid carcinoma (PTC) diagnosed [4]. TNs < 1 cm is now the most common incidental nodule [5]. Although PT microcarcinoma (PTMC) < 1 cm usually progresses slowly, cervical lymph node metastases may occur in 24–63% of patients [6]. Other studies have found that a node > 7 mm is an independent risk factor for lymph node metastasis (LNM) or distant metastasis [7]; therefore, it is necessary to accurately diagnose the benign and malignant TNs in the early stages and formulate corresponding treatment measures.

Ultrasound plays an essential role as a screening tool in TN management [8]. Ultrasonography is the most common, useful, safe, and cost-effective method of thyroid pathological imaging [9]. The widespread use of high-frequency ultrasound machines is the main reason that the prevalence of thyroid cancer appears to be increasing [1]. The American College of Radiology (ACR) published a white paper in 2017 on thyroid imaging reporting and data systems (TI-RADS), in which different risk levels for TNs were proposed and helpful for radiologists to standardize the description of the lesions. The TNs were graded using scores for the following sonographic features: composition, shape, echo, margins, and calcifications [10]. In their diagnoses, doctors are influenced by experience when using ultrasound, and senior doctors have higher diagnostic accuracy than junior doctors [11]. For TNs < 1 cm, the diagnosis using ultrasound images is more dependent on the doctor’s experience. ACR guidelines state that FNA evaluation for TNs < 1 cm is not recommended regardless of the ultrasound characteristics of the lesions [10]. There are studies on the diagnosis of TNs < 1 cm using the features in conventional ultrasound and contrast-enhanced modes [12]. Some researchers have suggested that the size of the nodule will affect the diagnosis of benign and malignant, and have found that the incidence of malignant tumors decreased with an increase in diameter, that the size of the nodule is inversely proportional to the risk of a malignant tumor, and that nodule size should not be regarded as an independent risk factor [13,14]. Therefore, a better, noninvasive diagnostic and therapeutic approach was needed to assess the status of TNs < 1 cm and avoid missing important diagnostic information.

Radiomics is a medical imaging diagnostic tool that has emerged in recent years. It is applied in the clinic to improve the accuracy of a diagnosis, prognosis, and prediction by extracting image feature data. Radiomics has become more and more important in cancer research and is the bridge between imaging and patient medical individuation [15]. Many studies have combined radiomics with clinical characteristics to predict the lymph node status in patients, the heterogeneity of the tumor, or the prognosis for a tumor patient [16,17,18]. Some studies have suggested that the diagnosis of TN using a “Rad-score” is better than that of primary doctors using ACR TI-RADS [19]. Studies are limited on whether ultrasound radiomics techniques can help distinguish the benign and malignant TNs < 1 cm.

The aim of the present study was to establish a model-visualized nomogram comprising radiomics combined with ACR TI-RADS to improve the diagnostic accuracy of TNs < 1 cm.

## 2. Materials and Methods

### 2.1. Patients and Groups

The present study comprised TN patients admitted to the Affiliated Hospital of Nantong University (training set) from March 2021 to March 2022 and to the Jiangsu Provincial Hospital of Integrated Traditional Chinese and Western Medicine (test set) from September 2017 to March 2022. The training set comprised 156 patients and the external test set 116 patients. This retrospective study waived informed consent and was approved by the hospital review board. The study inclusion criteria were as follows: (1) the maximum diameter of the target nodule was <1 cm, (2) pathological results were obtained by FNA or surgical resection, (3) ultrasonography was conducted within 2 weeks of FNA or surgery, and (4) the clinical and ultrasound data on the patients were complete. The exclusion criteria were as follows: (1) biopsy or resection had been conducted before ultrasonography, (2) thyroid inflammation, (3) other neoplastic diseases, (4) poor quality of ultrasound images, or (5) unclear pathological results. Figure 1 shows the patient pathways that two hospitals recruited according to the criteria of 272 patients. From the Affiliated Hospital of Nantong University, 122 women and 34 men were divided into the training set. The mean age was 45.64 ± 12.00 years and ranged from 18 to 76 years. From the Jiangsu Provincial Hospital of Integrated Traditional Chinese and Western Medicine, 92 women and 24 men were divided into the test set. The mean age was 47.47 ± 11.20 years and ranged from 20 to 77 years. More detailed selection steps can be obtained in Appendix A.

### 2.2. Clinical Data and Ultrasound Images

Baseline clinicopathological information comprised sex, age, ultrasonographic appearance (largest nodule diameter, location), and pathological results. The included ultrasound images were collected using the Esaote Mylab Class_C (Esaote Group, Genoa, Italy, shallow probe) and the HITACHI EUB-8500 (Hitachi, Tokyo, Japan, shallow probe) ultrasound equipment. The dynamic images and the maximum-diameter tomographic images of the nodules when scanning the lesions were retained and were stored in the picture archiving and communication system workstation for later measurements and evaluations by the doctors. The TI-RADS scoring items comprise composition, echo, shape, margin, and calcification. Two senior radiologists with >8 years of work experience from each hospital measured and scored the lesions in all patients. The four doctors were unaware of other assessments and pathological results, and any differences were resolved through negotiation to ensure the reliability of the data. Any histological and pathological findings of unclear significance were discarded, and only accurate and undisputed reports were used.

### 2.3. Region of Interest Segmentation and Radiomics Feature Extraction

The ultrasound images of all patients from the two different hospitals were uniformly formatted. Two radiologists (Doctor A and Doctor B) independently reviewed the ultrasound images. For each patient, a two-dimensional ultrasound image was selected that showed the maximum diameter of the nodule. The selected images were confirmed for consistency by the two radiologists. The open-source software ITK-SNAP (version 3.8.0; http://www.itksnap.org, accessed on 1 January 2022) was used to delineate and save the region of interest (ROI) of the target nodule on the selected ultrasound image. The features of the image, such as the first-order, morphology, gray histogram, and wavelet transform, were extracted using the extension package SlicerRadiomics in the open-source software 3D-Slicer (version 4.13.0; https://www.slicer.org, accessed on 20 March 2022), and the data were then stored. Intraclass and interclass correlation coefficients (ICC) were used to assess the reproducibility of extracted radiomics features. The ultrasound images of the nodules of 50 patients were randomly selected. Doctors A and B independently delineated the ROIs of the selected images. Doctor A delineated the lesion image again 2 weeks later. ICC analysis was conducted on the three sets of data so that the intra-observer and inter-observer reproducibility of features extracted by the two radiologists could be assessed. Features with ICCs > 0.8 were considered excellent and reliable.

### 2.4. Feature Selection and Rad-Score Building

Redundancy analysis was used on the feature data extracted from the training set. First, a normality test was conducted, and Pearson correlation analysis was used for normal features and Spearman correlation analysis was used for non-normal features. Redundant features were deleted with correlation coefficients > 0.9. The least absolute shrinkage and selection operator (LASSO) logistics regression algorithm was used for the remaining features, and five cross-validations were used to select the optimal penalty coefficient lambda. The radiomics features with non-zero coefficients in the training set were obtained, and these non-zero coefficients were weighted to form a Rad-score formula. This formula was used to calculate the Rad-score for each patient in the training and test sets, and the association between the Rad-score and TNs was assessed using the Mann–Whitney U-test.

### 2.5. Constructing Ultrasound Radiomics Nomogram

Taking *p* < 0.05 as the retention criterion and conducting a t-test for continuous variables and a chi-squared test for categorical variables, a univariate analysis of the clinical factors (including ACR TI-RADS score items) was conducted to determine risk factors related to benign and malignant nodules. A multivariate logistics regression analysis was conducted on the Rad-score and independent risk factors to determine the significant variables for the evaluation of TNs. Based on the variables selected using multivariate logistics regression analysis of the training set, a radiomics nomogram was established. Additional predictive models were established based on clinical factors and radiomics scores, respectively, to compare performance differences between models.

### 2.6. The Performance of Ultrasound Radiomics Nomogram

The nomogram calculated the total score of each patient by entering the corresponding independent risk factor scores, and the best Nomo-score cutoff value was assessed by maximizing the Youden index. The calibration curve and Hosmer–Lemeshow test were used to evaluate the correction effect of the nomogram on the training and test sets. The discriminative performance of the nomogram was evaluated using receiver operating characteristic (ROC) curves. Differences in the area under the curve (AUC) among the three models were compared using the DeLong test. A decision curve analysis (DCA) and clinical impact curve (CIC) were used to evaluate the clinical application value of the nomogram at different thresholds.

### 2.7. Statistical Analyses

R version 4.0.5 and SPSS version 24.0 (IBM Inc., Armonk, NY, USA) were used for statistical analyses. MedCalcm version 20.1 was used to draw and analyze the ROC curves. All statistical significance levels were two-sided, with *p*-values < 0.05 and a 95% confidence interval (CI). R and the associated step algorithm are provided in Appendix A.

## 3. Results

### 3.1. Clinical Information

The radiomics flow chart is shown in Figure 2. Clinical information on the training set and external independent validation set is summarized in Table 1. Except for the three features of shape, margin, and echogenicity that were significantly different in two datasets, the remaining clinicopathological features were not significantly different. The pathological results of the training and test datasets were not significantly different (*p* = 0.651). In the training set, using univariate analysis for each clinical factor, four factors were significantly different in the benign and malignant groups—margin, shape, calcification, and TI-RADS classification (Table 2); therefore, multivariate logistic regression analysis was used to construct a clinical model for predicting benign or malignant nodules based on these four predictive risk factors. It should be noted that TI-RADS classification was excluded because there was no significant difference in the regression analysis data (*p* = 0.1298).

### 3.2. Establishment of Radiomics Score

Radiomic features were extracted from the two-dimensional images of each patient using 3D-Slicer with 464 features in each group. After random sampling for ICC analysis, the results showed that intragroup ICCs ≥ 0.8 was 92.6% (430), and intergroup ICCs ≥ 0.8 was 89.6% (416). Both have good repeatability of intra-observer and inter-observer operations. As Appendix A shows, ten features with non-zero coefficients were screened for the construction of the radiomics score using LASSO regression on the training set. The relevant features, weighting coefficients, and specific formulas are provided in the Appendix A. The Rad-score of the malignant nodule group was significantly higher than that of the benign group in both training and test sets (Table 2).

### 3.3. Constructing and Evaluating Nomogram

Multivariate logistics regression analysis showed that four factors were significant—margin, calcification, shape, and Rad-score—as independent risk predictors for TN patients (Table 3). A radiomics nomogram was constructed with these four predictors (Figure 3a). The calibration curves show the calibration effect of the nomogram of the training set and test sets in predicting benign and malignant TNs (Figure 3b), and the Hosmer–Lemeshow test was used to observe the *p*-values of the two calibration curves (training set: 0.588; test set: 0.435).

Table 4 indicates the performance of the radiomics nomogram in predicting benign and malignant TNs. The optimal threshold of the nomogram score for judging benign and malignant TNs was determined by maximizing the Youden index of 0.486 (Figure 4). The AUC value of the nomogram in the training set was 0.866 (95% confidence interval (CI), 0.802–0.915), and the AUC value of the nomogram in the test set was 0.866 (95% CI, 0.790–0.922), both of which reflect that the nomogram has good discrimination ability. The clinical and Rad-score models were performed separately for comparison (Figure 4). In the training set (Figure 4c), the nomogram and clinical model AUCs were 0.866 and 0.795 (*p* = 0.0019), respectively, the nomogram and Rad-score model AUCs were 0.866 and 0.774 (*p* = 0.0053), respectively, and the Rad-score and clinical model AUCs were 0.774 and 0.795 (*p* = 0.6748), respectively. In the test set (Figure 4d), the nomogram and clinical model AUCs were 0.866 and 0.783 (*p* = 0.0099), respectively, the nomogram and Rad-score model AUCs were 0.866 and 0.740 (*p* = 0.0006), respectively, and the Rad-score and clinical model AUCs were 0.740 and 0.783 (*p* = 0.4835), respectively. It was observed that the radiomics nomogram had better discrimination than the clinical model and the Rad-score model (Table 5). In addition, the waterfall chart was predicted with a threshold score (0.486), showing the predictions of the test set patients separately (Figure 5).

Figure 6a shows the DCA curves of all patients under the different models. When thresholds were between 0.2 and 0.8, using the nomogram to diagnose TNs was more accurate than using the clinical model or Rad-score alone. Figure 6b shows the CIC of the radiomics nomogram and demonstrates the high clinical efficacy of the nomogram. When the risk threshold probability was >70% of the predicted score probability value, the malignant high-risk population and the actual malignant population had a high degree of matching.

## 4. Discussion

The present study focused mainly on the identification of benign and malignant TNs < 1 cm. In the training set, univariate analysis was conducted on clinical factors, including TI-RADS score items, to acquire significant variables, then combined them with the radiomics features selected by LASSO regression. These variables were then subjected to multivariate logistics regression to obtain a radiomics nomogram, which had excellent performance in predicting benign and malignant TNs < 1 cm.

Imaging is crucial for early cancer detection, and radiomics is helped to build powerful classifier models to improve cancer screening and early detection [20]. Constructing a Rad-score by screening relevant radiomics features using LASSO has been widely used in the research on related ultrasound radiomics [21,22,23]. LASSO finally screened out 10 features with non-zero coefficients related to the TNs < 1 cm. The sphericity feature had the largest weight coefficient. This feature is similar to the results of Dorota et al. [24] in evaluating the relationship between TN shape and malignant nodules.

In the training dataset, four clinical features—margins, shape, calcification, and TI-RADS grade—were selected using univariate analyses to be meaningful in distinguishing between benign and malignant TNs < 1 cm (*p* < 0.001); however, the subsequent multivariate logistics regression analysis showed insignificant differences in TI-RADS classification (*p* = 0.1298). Thus, this factor was excluded. Constructing a nomogram with clinical features and radiomics scores showed positive performance in the training and test sets. Several studies have combined radiomics features with clinical factors to construct a model to judge disease status [25,26]. Radiomics not only assesses the image features of the patients but also takes other factors into account. The diagnosis and treatment of patients are more individualized and comprehensive and used to resolve clinical issues [27].

Some studies on ACR TI-RADS have indicated that the diagnostic accuracy of TNs corresponds highly with the experience of the radiologists and will be more dependent on their overall experience [28]. Although guidelines recommend follow-up visits for TNs < 1 cm, 72% of PTMC patients choose surgery rather than follow-up [29]. In addition, there remains a small number of significant cases of PTMC that exhibit more biologically aggressive features, such as lymph node involvement and early metastases [30]. Ye et al. [31] have shown that ~39.1% of patients with PTMC have developed central lymph node metastases. Other studies have used 5 mm as the cutoff value to observe the differences between PTMC and pathological features [32]. The results of the above studies may concern many patients, intervening in micronodules with high-risk features.

Zhang et al. [33] have shown that the sensitivity of diagnosing TNs with ultrasound alone was 77.14%, the specificity was 65.79%, and the AUC was 0.728. Ultrasound-guided FNA, studied by Gao et al. [34] for the diagnosis of TNs < 1 cm, showed excellent diagnostic performance (sensitivity, specificity, and AUC: 98.8%, 90.5%, and 0.947, respectively); however, it was an invasive method. In the present study, the sensitivity, specificity, and AUC of the nomogram constructed using clinical factors combined with radiomics were 78.87%, 85.88%, and 0.866, respectively. The results of the present study indicated that the nomogram may be a better choice than ultrasonography alone or FNA invasive examination. In addition, DCA and CIC curves are often used to evaluate the clinical value of various models [35,36], showing the clinical reliability of the nomogram, and compared with the clinical model and Rad-score, the nomogram has a greater improvement. The cutoff value corresponding to the maximum Youden value can also help in clinical evaluation. When the Nomo-score is >0.486, the TN < 1 cm is at high risk, and the patient can be given additional examination or treatment.

This study had some limitations. First, it was a retrospective study; therefore, there were some selection biases and imbalances in the data. The study included pathological results of TN nodules, all graded above TR2, because FNA or surgical resection for benign performance could not be implemented, which may have affected the results. More prospective studies are needed to verify the feasibility of using the nomogram in the future. Second, radiomics features were extracted on two-dimensional ultrasound images. Compared with three-dimensional images, some disease information may be missed using two-dimensional images. In the future, software and hardware optimization may be required to observe three-dimensional ultrasound images and obtain more information on diseases. In addition, the ultrasound examination and the delineation of the radiomics ROI are manual, which will inevitably lead to some mistakes; therefore, two experienced doctors were used in the present study to conduct the operation and consistent tests to avoid adverse effects as much as possible.

## 5. Conclusions

The present study proposed an ultrasound radiomics nomogram based on ACR TI-RADS for noninvasively predicting benign and malignant TNs < 1 cm, which proved to be superior to using the radiomics model alone or ACR TI-RADS.

## Figures and Tables

**Figure 1 cancers-14-04826-f001:**
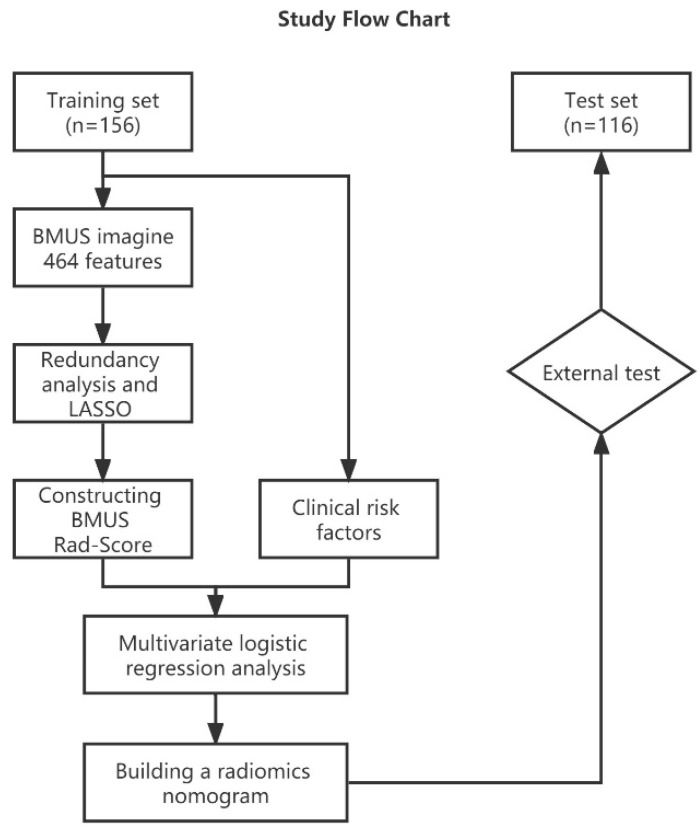
Method flow chart of the present study.

**Figure 2 cancers-14-04826-f002:**
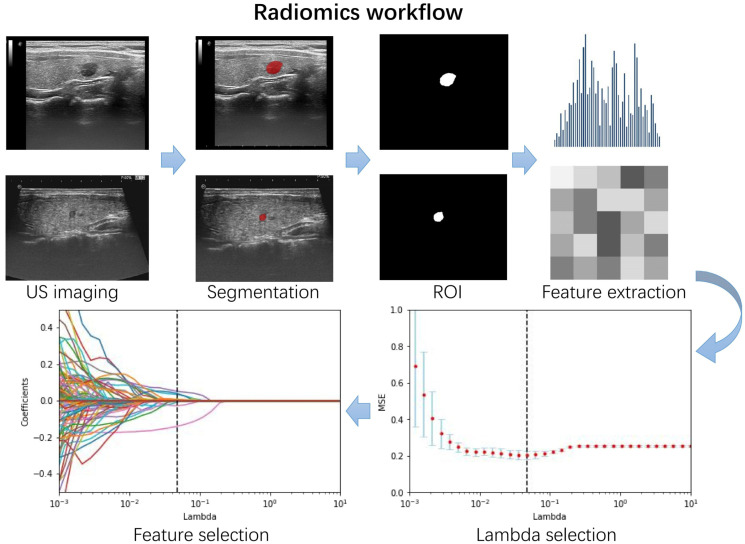
Flow chart showing the process of radiomics. One image from each of the two hospitals was selected as a demonstration. Region of interest (ROI) segmentation was conducted on the largest diameter of the nodule. Radiomics features were extracted from ROIs, including features such as shape, grayscale, texture, and wavelets. The optimal penalty coefficient lambda in the least absolute shrinkage and selection operator (LASSO) model was generated using five-fold cross-validation and a minimum criteria procedure. Ultrasound image features were selected using LASSO.

**Figure 3 cancers-14-04826-f003:**
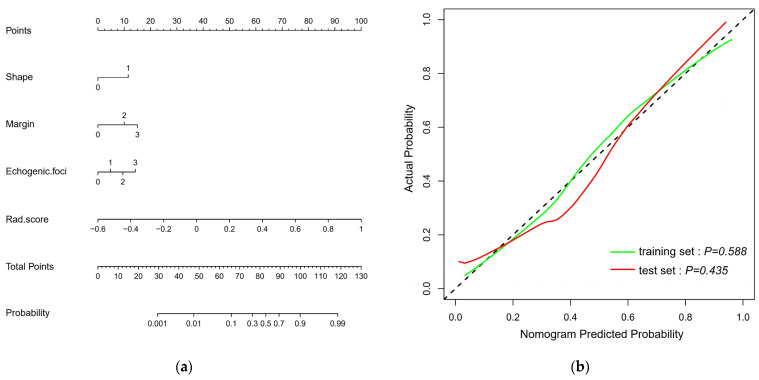
(**a**) A radiomics nomogram for predicting benign and malignant thyroid nodules (TNs) < 1 cm. (**b**) The green and red lines represent the calibration curves of the nomogram in the training and test sets, respectively, and the dashed black line represents the ideal case.

**Figure 4 cancers-14-04826-f004:**
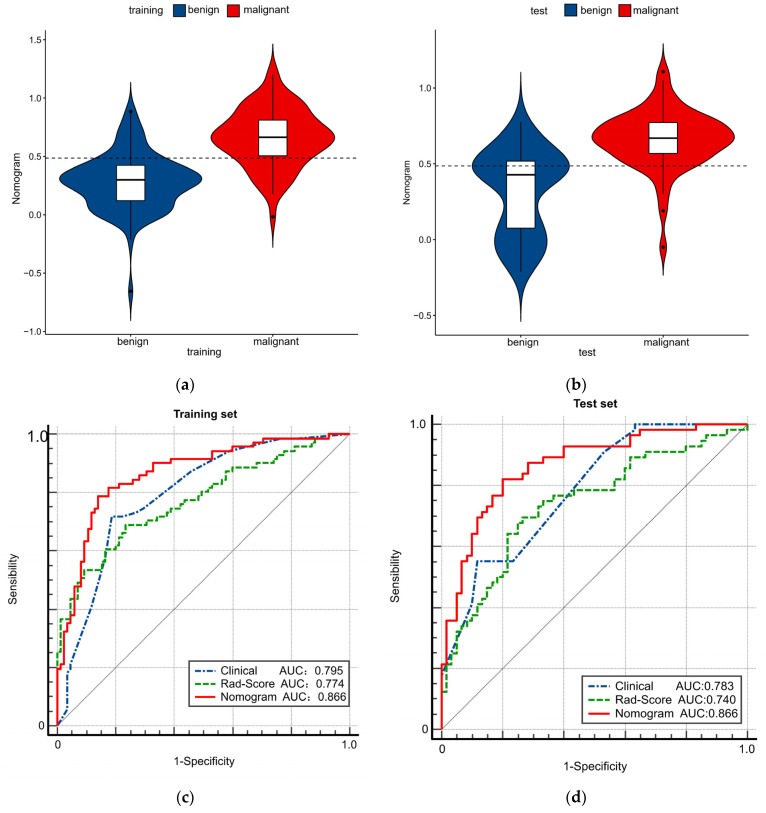
The role of radiomics nomogram in predicting benign and malignant thyroid nodules (TNs). The violin plot shows the data distribution, including a box plot. The cutoff value for the Nomo-score is 0.486 in both training (**a**) and test (**b**) sets. The receiver operating characteristic curves of the clinical model, the radiomics model, and the nomogram are shown in the training (**c**) and test (**d**) sets, respectively.

**Figure 5 cancers-14-04826-f005:**
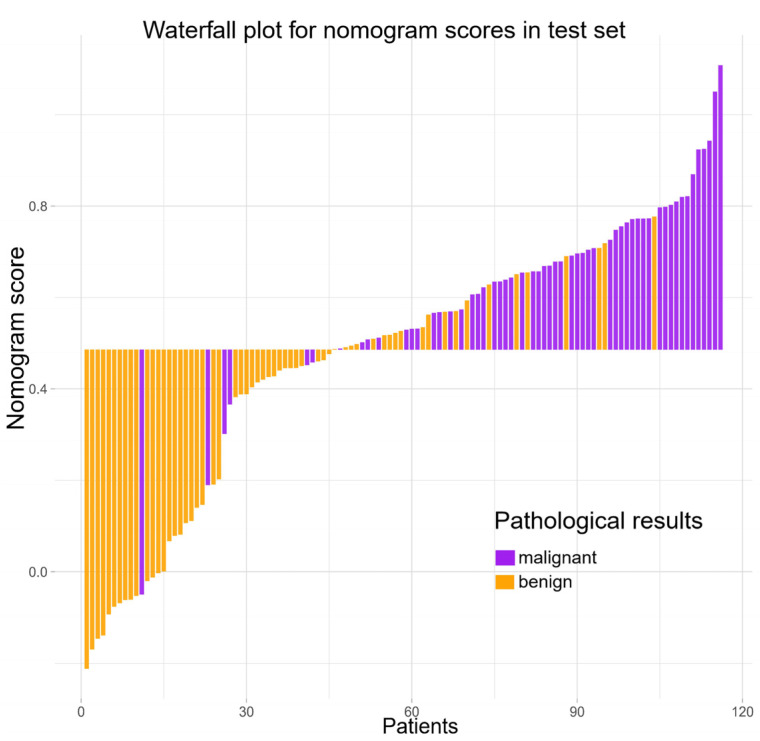
The waterfall chart shows the prediction of the test set patients (*n* = 116) at a threshold of 0.486, as well as the Nomo-score and pathological result for each patient. The two colors represent different pathological results—purple for malignant nodules and orange for benign nodules.

**Figure 6 cancers-14-04826-f006:**
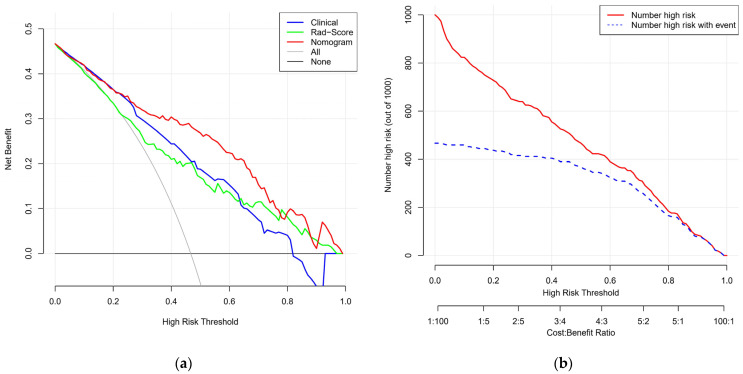
(**a**) Decision analysis curve (DCA) showing the role of different models in predicting benign and malignant thyroid nodules (TNs). The vertical axis represents the net benefit, and the horizontal axis represents different risk thresholds. (**b**) The clinical impact curve (CIC) of the radiomics nomogram for predicting risk stratification in 1000 people, with predictive power at different threshold probabilities.

**Table 1 cancers-14-04826-t001:** Clinicopathological characteristics in the training and the test datasets.

Characteristic	Training Dataset No. (%)	Test Dataset No. (%)	*p*-Value
Sex			0.826
male	34 (21.8)	24 (20.7)	
female	122 (78.2)	92 (79.3)	
Age, mean ± SD, years	45.64 ± 12.00	47.47 ± 11.20	0.201
Pathology			0.651
benign	85 (54.5)	60 (51.7)	
malignant	71 (45.5)	56 (48.3)	
Tumor location			0.225
left	84 (53.8)	71 (61.2)	
right	72 (46.2)	45 (38.8)	
Tumor size, mean ± SD, mm	7.30 ± 1.73	7.41 ± 1.66	0.609
TI-RADS level			0.903
TR 3	1 (0.6)	6 (5.2)	
TR 4	36 (23.1)	21 (18.1)	
TR 5	119 (76.3)	89 (76.7)	
Composition			0.119
solid	154 (98.7)	111 (95.7)	
mixed cystic and solid	2 (0.3)	5 (4.3)	
Echogenicity			<0.001
hyperechoic or isoechoic	7 (4.5)	21 (18.1)	
hypoechoic	139 (89.1)	93 (80.2)	
very hypoechoic	10 (6.4)	2 (1.7)	
Shape			0.005
wider-than-tall	90 (57.7)	47 (40.5)	
taller-than-wide	66 (42.3)	69 (59.5)	
Margin			<0.001
smooth or ill-defined	82 (52.6)	23 (19.8)	
lobulated or irregular	65 (41.7)	83 (71.6)	
extra-thyroidal extension	9 (5.7)	10 (8.6)	
Echogenic foci			0.206
none or large comet-tail artifacts	68 (43.6)	64 (55.2)	
macrocalcifications	14 (9.0)	9 (7.8)	
peripheral (rim) calculations	6 (3.8)	6 (5.1)	
punctate echogenic foci	68 (43.6)	37 (31.9)	
BMUS Rad-score, median (interquartile range)	0.40 (0.32–0.52)	0.41 (0.30–0.51)	0.358

Note: The capital initials in the first column of the table represent the variables included in the study that correspond to the *p*-value. No capitalization indicates detailed sub-variables that correspond to frequencies and percentages. Abbreviations: TI-RADS: thyroid imaging, reporting and data system; BMUS, B-model ultrasound; Rad-score, radiomics score. Data are the number of patients and percentage if not specified.

**Table 2 cancers-14-04826-t002:** Predictors for TN status in the training and the test datasets.

Characteristic	Training Dataset No. (%)			Test Dataset No. (%)		
	Malignant	Benign	*p*-Value	Malignant	Benign	*p*-Value
Sex			0.078			0.788
male	20 (28.2)	14 (16.5)		11(19.4)	13 (21.7)	
female	51 (71.8)	71 (83.5)		45 (80.6)	47 (78.3)	
Age, mean ± SD, years	44.01 ± 11.57	47.00 ± 12.25	0.122	46.13 ± 10.30	48.73 ± 11.92	0.211
Tumor location			0.372			0.782
left	41 (57.7)	43 (50.6)		35 (62.5)	36 (60.0)	
right	30 (42.3)	42 (49.4)		21 (37.5)	24 (40.0)	
Tumor size, mean ± SD, mm	7.20 ± 1.79	7.39 ± 1.69	0.5	7.76 ± 1.68	7.08 ± 1.59	0.027
TI-RADS level			<0.001			<0.001
TR 3	0	1 (1.2)		0	6 (10.0)	
TR 4	4 (5.6)	32 (37.6)		3 (5.4)	18 (30.0)	
TR 5	67 (94.4)	52 (61.2)		53 (94.6)	36 (60.0)	
Composition			0.193			0.196
solid	71 (100)	83 (97.6)		55 (98.2)	56 (93.3)	
mixed cystic and solid	0	2 (2.4)		1 (1.8)	4 (6.7)	
Echogenicity			0.077			0.046
hyperechoic or isoechoic	3 (4.2)	4 (4.7)		5 (8.9)	16 (26.7)	
hypoechoic	60 (84.5)	79 (92.9)		50 (89.3)	43 (71.7)	
very hypoechoic	8 (11.3)	2 (2.4)		1 (1.8)	1 (1.6)	
Shape			0.01			0.076
wider-than-tall	33 (46.5)	57 (67.1)		18 (32.1)	29 (48.3)	
taller-than-wide	38 (53.5)	28 (32.9)		38 (67.9)	31 (51.7)	
Margin			<0.001			<0.001
smooth or ill-defined	26 (36.6)	56 (65.9)		1 (1.8)	22 (36.7)	
lobulated or irregular	37 (52.1)	28 (32.9)		45 (80.4)	38 (63.3)	
extra-thyroidal extension	8 (11.3)	1 (1.2)		10 (17.8)	0	
Echogenic foci			0.007			0.001
none or large comet-tail artifacts	24 (33.8)	44 (51.8)		30 (53.6)	34 (56.7)	
macrocalcifications	3 (4.2)	11 (12.9)		1 (1.8)	8 (13.3)	
peripheral (rim) calculations	3 (4.2)	3 (3.5)		0	6 (10.0)	
punctate echogenic foci	41 (57.8)	27 (31.8)		25 (44.6)	12 (20.0)	
BMUS Rad-score, median (interquartile range)	0.50 (0.37–0.64)	0.36 (0.30–0.42)	<0.001	0.46 (0.37–0.62)	0.33 (0.25–0.42)	<0.001

Notes: The capital initials in the first column of the table represent the variables included in the study that correspond to the *p*-value. No capitalization indicates detailed sub-variables that correspond to frequencies and percentages. Data are number of patients and percentage if not specified.

**Table 3 cancers-14-04826-t003:** Risk factors for malignant nodules in the training cohort.

Intercept and Variable	Clinical Model			BMUS Radiomics Nomogram		
	β	Odds Ratio (95% CI)	*p*-Value	β	Odds Ratio (95% CI)	*p*-Value
Intercept	−2.606			−5.867		
Shape	1.493	4.452 (2.004–9.890)	<0.001	1.489	4.431 (1.799–10.913)	0.001
Margin	0.842	2.322 (1.610–3.349)	<0.001	0.646	1.909 (1.276–2.854)	0.002
Echogenic foci	0.595	1.813 (1.364–2.410)	<0.001	0.614	1.847 (1.342–2.542)	<0.001
TI-RADS level	NA	NA	NA	NA	NA	NA
BMUS Rad-score	NA	NA	NA	8.079	2.243 (1.581–3.184)	<0.001

Abbreviations: TI-RADS: thyroid imaging reporting and data system; BMUS, B-model ultrasound; Rad-score, radiomics score; AUC: area under the curve.

**Table 4 cancers-14-04826-t004:** Performance of different models for evaluating thyroid nodule (TN) status.

Variable	Clinical Model (95% CI)		Rad-Score (95% CI)		BMUS Radiomics Nomogram (95% CI)	
	Training Cohort	Test Cohort	Training Cohort	Test Cohort	Training Cohort	Test Cohort
Cutoff value	0.587	0.587	0.421	0.421	0.486	0.486
AUC	0.795 (0.723–0.855)	0.783 (0.697–0.854)	0.774 (0.700–0.837)	0.740 (0.651–0.817)	0.866 (0.802–0.915)	0.866 (0.790–0.922)
PLR	4.00 (2.57–6.24)	3.79 (2.25–6.41)	3.62 (2.35–5.57)	2.93 (1.84–4.67)	6.70 (3.83–11.73)	8.88 (3.82–20.64)
NLR	0.27 (0.17–0.43)	0.28 (0.17–0.47)	0.31 (0.20–0.47)	0.36 (0.22–0.56)	0.16 (0.09–0.29)	0.20 (0.11–0.34)
Sensitivity, %	78.26 (66.69–87.29)	77.19 (64.16–87.26)	75.71 (63.99–85.17)	73.21 (59.70–84.17)	85.71 (75.29–92.93)	82.26 (70.47–90.80)
Specificity, %	80.46 (70.56–88.18)	79.66 (67.17–89.02)	79.07 (68.95–87.10)	75.00 (62.14–85.28)	87.21 (78.26–93.44)	90.74 (79.70–96.92)
PPV, %	76.06 (67.08–83.20)	78.57 (68.46–86.10)	74.65 (65.66–81.93)	73.21 (63.17–81.33)	84.51 (75.70–90.52)	91.07 (81.45–95.95)
NPV, %	82.35 (74.67–88.08)	78.33 (68.79–85.57)	80.00 (72.28–85.98)	75.00 (65.51–82.57)	88.23 (80.77–93.05)	81.67 (72.14–88.46)
Diagnostic accuracy, %	79.49 (72.29–85.53)		73.08 (65.40–79.86)	74.14 (65.18–81.82)	86.54 (80.16–91.47)	86.21 (78.57–91.91)
TP	51	31	43	36	54	43
TN	67	46	71	46	73	48
FP	18	14	14	14	12	12
FN	20	25	28	20	17	13
Brier score	0.181	0.181	0.186	0.209	0.144	0.153

AUC, the area under the receiver operating characteristic curve; PLR, positive likelihood ratio; NLR, negative likelihood ratio; PPV, positive predictive value; NPV, negative predictive value; TP, true positive; TN, true negative; FP, false positive; FN, false negative.

**Table 5 cancers-14-04826-t005:** Comparison of area under the curve (AUC) among different models.

	Training Cohort (95% CI)		Test Cohort (95% CI)	
	AUC	*p*-Value	AUC	*p*-Value
Clinical vs. Rad-score	0.795 vs. 0.774	0.6748	0.783 vs. 0.740	0.4835
Clinical vs. Nomogram	0.795 vs. 0.866	0.0019	0.783 vs. 0.866	0.0099
Rad-score vs. Nomogram	0.774 vs. 0.866	0.0053	0.740 vs. 0.866	0.0006

Notes: The Delong test compared the models from the training cohort and the test cohort. Abbreviations: Rad-score, radiomics score; AUC: area under the receiver operating characteristic curve.

## Data Availability

Original comments presented in the research report are contained in the article/Appendix A. For further inquiries, the corresponding authors may be contacted directly.

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
