# Peer review of "Ultrasound Radiomics Nomogram to Diagnose Sub-Centimeter Thyroid Nodules Based on ACR TI-RADS"

_cancers, 2022, doi:10.3390/cancers14194826_

Round 1

Reviewer 1 Report

The article was informative and well designed.

Suggest adding diagnostic accuracy measures: TP, TN, FP, FN, and Brier error

Is clinical outcome variables existed to test the nomogram or generate another one for predicting progression in cancer nodules

Suggest adding supplementary R codes used for the readers

Author Response

Response to Reviewer 1 Comments

Point 1: Suggest adding diagnostic accuracy measures: TP, TN, FP, FN, and Brier error

Response 1: Thank you for your suggestion. We have included relevant suggestions in Table 4.

Point 2: Is clinical outcome variables existed to test the nomogram or generate another one for predicting progression in cancer nodules

Response 2: Thank you for your question. Shape, Margins, and Echogenic foci features can be used as clinical variables to test Nomogram in the diagnosis of sub-centimeter thyroid nodules by BMUS. We are sorry that we have not done prospective study on Nomogram. In the future, we will further investigate whether there are clinical variables that can predict the progression of cancer nodules.

Point 3: Suggest adding supplementary R codes used for the readers

Response 3: Thanks for your suggestion, we will provide the relevant R code in the supplementary file.

Reviewer 2 Report

The authors have presented a paper with the title "Ultrasound radiomics nomogram diagnose sub-cetimeter thyroid nodules based on ACR TI-RADS", and have proposed an ultrasound radiomics nomogram for non-invasively predict benign and malignant sub-centimeter thyroid nodules.

The paper is well structured but need improvement. There are English language issues that need to be corrected. Line 16, abstract, line 35 (TN have increased?? The incident has increased!), lines 41, 47-48, etc.. (Please request English editing).

The tables could and should be improved to make it easier to interpret. 

Line 41: the percentage 20% to 90% seems to be too high....

Is reference 6 ok??

Concerning the methods used I do not feel comfortable to comment since it is not my area of expertise.

In the tables where percentages are included, it should be stated (eg.:  n (%) - 34 (21.8)).

In table 1 and 2, "gender" should be substituted by "sex".

Line 287, which factors??

Description in lines 175 to 178 do not corresponde to results in table 1.

The study have some limitations already stated by the authors.

Author Response

Response to Reviewer 2 Comments

Point 1: The paper is well structured but need improvement. There are English language issues that need to be corrected. Line 16, abstract, line 35 (TN have increased?? The incident has increased!), lines 41, 47-48, etc.. (Please request English editing).

Response 1: Thank you for pointing out the English language issue with our article. Our article has been resubmitted with English editing.

Point 2: The tables could and should be improved to make it easier to interpret. 

Response 2: Thank you for your suggestion. Due to the large number of variables included in the ACR standard, the content of the table is complicated during statistical analysis. We are very sorry for this. We explain the content further below the Table 1 and 2.

Point 3: Line 41: the percentage 20% to 90% seems to be too high.

Response 3: We are very sorry for the citation errors caused by our carelessness, and thank you for your careful correction. The reference data is 52.8%.(Has been corrected in line 48)

Point 4: Is reference 6 ok?

Response 4: We are very sorry for the mistake we made. Reference 6 was not scrutinized when cited, thank you for your careful review, it has now been corrected.(Modified on page 14, lines 387-388)

Point 5: Concerning the methods used I do not feel comfortable to comment since it is not my area of expertise.

Response 5: Thank you for your review of our manuscript. The methodology of this article can be summarized as image collection, ROI segmentation, feature extraction, feature screening, statistical analysis, and modeling. The method is in line with the radiomics process. You can refer to our previous work, the two studies have similarities in method (DOI: 10.3389/fonc.2021.709339).

Point 6: In the tables where percentages are included, it should be stated (eg.:  n (%) - 34 (21.8)).

Response 6: We have modified the first row of Table 1 as you suggested.

Point 7: In table 1 and 2, "gender" should be substituted by "sex".

Response 7: Thank you for your suggestion. Table 1 and 2 have been corrected as you suggested.

Point 8: Line 287, which factors?

Response 8: Thank you for your careful review. Among the ten radiomics features screened by LASSO, the Sphericity feature has the largest weight coefficient. The original text was written as Shape due to carelessness. The corresponding reference is also the feature of Sphericity.(Modified on page 12, lines 297-298)

Point 9: Description in lines 175 to 178 do not corresponde to results in table 1.

Response 9: We're sorry that we didn't express the results clearly enough, we re-described this part of the results. (Modified on page 5, lines 185-189)

Point 10: The study have some limitations already stated by the authors.

Response 10: Thank you very much for your review. In the future ,we hope to find ways to avoid these limitations.

Reviewer 3 Report

It is a very well written and intriguing manuscript. Well done!

·       As you mention on your limitations you used two-dimensional US images. Is there a purpose of not using  three-dimensional US images?

·       Various radiomic methods exist. Why did you choose the LASSO Logistic regression algorithm?

·       LINE 60-61 The ACR guidelines believe that FNA evaluation for nodules smaller than 1 cm is not recommended regardless of the ultrasound characteristics of 62 the lesions (12)à please site ACR guidelines rather than ATA.

Author Response

Response to Reviewer 3 Comments

Point 1: As you mention on your limitations you used two-dimensional US images. Is there a purpose of not using three-dimensional US images?

Response 1: Thank you for your question. We also want to use three-dimensional images, but most ultrasound machines in the country can only get two-dimensional images. At present, our hospital only have a technology for breast examination can collect three-dimensional images(automated breast volume scanner, ABVS), other ultrasonic machines can not collect three-dimensional images. In the future, we intend to cooperate with ultrasonic machine manufacturers, hoping to develop volume imaging for thyroid examination, so that we can obtain three-dimensional images of the lesion.

Point 2: Various radiomic methods exist. Why did you choose the LASSO Logistic regression algorithm?

Response 2: Thank you for your question. The LASSO algorithm shrinks some coefficients and reduces others to exactly 0 via the absolute constraint. The radiomics extracted a large number of features. These features are not all useful, but also affect the simplification of the model. The Lasso algorithm reduced the high-dimensional problem brought by the radiomics features, and screened out important features. At the same time, it can directly observe the weight coefficient of important features. At present, LASSO has been used as an important algorithm for dimensionality reduction of the radiomics features. Our early work also used this algorithm (DOI: 10.3389/fonc.2021.709339).

Point 3: LINE 60-61. The ACR guidelines believe that FNA evaluation for nodules smaller than 1 cm is not recommended regardless of the ultrasound characteristics of 62 the lesions (12)à please site ACR guidelines rather than ATA.

Response 3: Thank you for your careful review, and the reference has been corrected as ACR. (Modified on page 2, line 69)
